# A Specific HPLC Method to Determine Residual HEPES in [^68^Ga]Ga-Radiopharmaceuticals: Development and Validation

**DOI:** 10.3390/molecules27144477

**Published:** 2022-07-13

**Authors:** Silvia Migliari, Maura Scarlattei, Giorgio Baldari, Claudia Silva, Livia Ruffini

**Affiliations:** 1Nuclear Medicine and Molecular Imaging Department, Azienda Ospedaliero-Universitaria di Parma, Via Gramsci 14, 43126 Parma, Italy; mscarlattei@ao.pr.it (M.S.); gbaldari@ao.pr.it (G.B.); lruffini@ao.pr.it (L.R.); 2Food and Drug Sciences Department, Parco Area delle Scienze 27/A, University of Parma, 43124 Parma, Italy; claudia.silva@unipr.it

**Keywords:** [^68^Ga]Ga-radiopharmaceuticals, HEPES, validation of HPLC method, quality controls

## Abstract

Background: Nowadays, in Nuclear Medicine, clinically applied radiopharmaceuticals must meet quality release criteria such as high radiochemical purity and radiochemical yield. Many radiopharmaceuticals do not have marketing authorization and have no dedicated monograph within European Pharmacopeia (Ph. Eur.); therefore, general monographs on quality controls (QCs) have to be applied for clinical application. These criteria require standardization and validation in labeling and preparation, including quality controls measurements, according to well defined standard operation procedures. However, QC measurements are often based on detection techniques that are specific to a certain chromatographic system. Several radiosyntheses of [^68^Ga]Ga-radiopharmaceuticals are more efficient and robust when they are performed with 2-[4-(2-hydroxyethyl)piperazin-1-yl] ethanesulfonic acid (HEPES) buffer, which is considered as an impurity to be assessed in the QC procedure, prior to clinical use. Thus, Ph. Eur. has introduced a thin-layer chromatography (TLC) method to quantify the HEPES amount that is present in [^68^Ga]Ga-radiopharmaceuticals. However, this is only qualitative and has proven to be unreliable. Here we develop and validate a new high-performance liquid chromatography (UV-Radio-HPLC) method to quantify the residual amount of HEPES in ^68^Ga-based radiopharmaceuticals. Method: To validate the proposed UV-Radio-HPLC method, a stepwise approach was used, as defined in the guidance document that was adopted by the European Medicines Agency (CMP/ICH/381/95 2014). The assessed parameters are specificity, linearity, precision (repeatability), accuracy, and limit of quantification. A range of concentrations of HEPES (100, 80, 60, 40, 20, 10, 5, 3 μg/mL) were analyzed. Moreover, to test the validity and pertinence of our new HPLC method, we analyzed samples of [^68^Ga]Ga-DOTATOC; [^68^Ga]Ga-PSMA; [^68^Ga]Ga-DOTATATE; [^68^Ga]Ga-Pentixafor; and [^68^Ga]Ga-NODAGA-Exendin-4 from different batches that were prepared for clinical use. Results: In the assessed samples, HEPES could not be detected by the TLC method that was described in Ph. Eur. within 4 min incubation in an iodine-saturated chamber. Our developed HPLC method showed excellent linearity between 3 and 100 μg/mL for HEPES, with a correlation coefficient (R^2^) for calibration curves that was equal to 0.999, coefficients of variation (CV%) < 2%, and percent deviation value of bias from 100% to 5%, in accordance with acceptance criteria. The intra-day and inter-day precision of our method was statistically confirmed and the limit-of-quantification (LOQ) was 3 μg/mL, confirming the high sensitivity of the method. The amount of HEPES that was detected with our developed HPLC method in the tested [^68^Ga]Ga-radiopharmaceuticals resulted well below the Ph. Eur. limit, especially for [^68^Ga]Ga-NODAGA-Exendin-4. Conclusions: The TLC method that is described in Ph. Eur. to assess residual HEPES in [^68^Ga]-based radiopharmaceuticals may not be sufficiently sensitive and thus unsuitable for QC release. Our new HPLC method was sensitive, quantitative, reproducible, and rapid for QCs, allowing us to exactly determine the residual HEPES amount in [^68^Ga]Ga-radiopharmaceuticals for safe patient administration.

## 1. Introduction

A radiopharmaceutical can be considered as a ‘’structured whole’’ that constitutes two parts: [1] chemistry, for which the concerns are chemical structure, receptor/target interactions, kinetics, synthesis, stability, etc., and [2] the regulatory and quality assurance aspects, which assure that these molecules possess proven quality, safety, and efficacy characteristics to be considered as medicinal products.

This qualification affects all drug molecules, regardless of their development stage (early discovery to approved drug) and source (natural product or synthetic). Purity assessment is perhaps most critical in discovery programs and whenever chemistry is linked with a biological and/or therapeutic outcome.

There have been many efforts to develop high-resolution imaging tools that can target specific disease hallmarks for patient stratification and treatment selection for a precision medicine. The novel radiopharmaceuticals have no marketing authorization and no dedicated monograph within Ph. Eur.; however, they must meet release criteria such as a high radiochemical yield (RCY) and chemical- and radio-chemical purity (RCP), according to QCs release [3,4,5,6,7].

Chemical purity refers to an element contains a single substance, without any other element tarnishing its standalone existence , regardless of the presence of radioactivity [7], while RCP refers to the percentage of the total radioactivity that is present in the desired chemical form in a radioactive pharmaceutical [8].

Chemical and radiochemical impurities may originate from the reaction reagents, incomplete labeling, breakdown of radiolabeled preparations with time as a result of instability, or through the introduction of inappropriate ingredients during the synthesis processes.

To separate (radio)chemical forms that are different from the radiopharmaceutical, it is important to have a validated and standardized analytical method ensuring clinical suitability and the adherence to well-defined standard operation procedures (SOPs) [4,5,9,10]. Separation chromatographic methods use techniques such as gas chromatography (GC), HPLC, or thin-layer chromatography (TLC) [11,12,13,14].

Indeed, validation and qualification activities are an integral aspect of the radiopharmacy routine [15], and guidance documents are published and continuously updated by ICH [16,17]; EDQM [18]; EANM; and IAEA/WHO [10,11,12,13,14,15,16,17,18,19,20].

We have synthesized PET radiopharmaceuticals such as [^68^Ga]Ga-DOTATOC, [^68^Ga]Ga-HBEDD-CC-PSMA [21]; [^68^Ga]Ga-DOTATATE [22]; [^68^Ga]Ga-DOTA-Pentixafor [23,24]; and [^68^Ga]Ga-NODAGA-Exendin-4 [25,26] using the automated synthesis module Scintomics GRP^TM^ (Scintomics GmbH, Fürstenfeldbruck, Germany) and tested them to assess all release criteria with a validated QCs system.

During the labeling process, different concentrations and volumes of buffer solutions may be used to adjust the acid pH of the eluate [^68^Ga]GaCl_3_ that is obtained from the ^68^Ge/^68^Ga generator [27].

HEPES (Figure 1) is a synthetic zwitterionic compound with strong buffering power that is widely used not only in chemical reactions, but also in cell cultures, thanks to its ability to maintain pH at a physiological value. However, despite being considered suitable for human use [28], due to the limited availability of data relating to its toxicity, this substance is considered a chemical impurity, and its residue must therefore be evaluated during the execution of the QCs of the radiopharmaceutical preparation before release.

Therefore, Ph. Eur. has introduced the following in the Monographs of Ga-68 labeled radiopharmaceuticals: Gallium (^68^Ga) Edotreotide injection Monograph 2482 [29], and Gallium (^68^Ga) PSMA-11 injection Monograph 3044 [30], an ad hoc test to detect the residue of HEPES.

This TLC method is based on a comparison of intensity color between the radiopharmaceutical and a reference solution of HEPES; therefore, it is only a qualitative method, and it does not provide information about the real residual amount of the impurity.

The purpose of the work is to develop and subsequently validate an HPLC analytical method for the quantification of residual HEPES in [^68^Ga]Ga-radiopharmaceuticals following the ICH Q2 (R1) guidelines [16,17].

## 2. Results

### 2.1. TLC Analysis of [^68^Ga]Ga-Radiopharmaceuticals

The HEPES reference sample of (200 μg/V_Injected_) in water was prepared according to Ph. Eur. [30,31], and a spot of each [^68^Ga]Ga-radiopharmaceutical was tested. After 4 min of incubation in iodine vapor the plate was observed, but no bright yellow spot was clearly visible (data not shown). This result was replicated for all the tested [^68^Ga]Ga-radiopharmaceuticals.

### 2.2. Results of Validation of the HPLC Method for the Quantification of HEPES in [^68^Ga]Ga-Radiopharmaceuticals

To validate the HPLC method, a calibration curve was generated from an average of five measurements containing 8 different concentrations of HEPES of between 3 and 100 μg/mL (Figure 2, Figure 3 and Figure 4).

The correlation coefficient of the calibration curve between the HEPES concentration and the peak area in the range of 3–100 μg/mL was an R^2^ value of 0.999. To illustrate the intra-day and inter-day precision of the method, five replicate determinations of HEPES were performed using three different concentrations across three consecutive days.

#### 2.2.1. Solvent Effect

The intensity of the peaks, as well as the retention time of HEPES, dissolved in different solutions, were found to be not significantly different from each other (Table 1); therefore, a matrix effect did not interfere in the HEPES affinity for the stationary phase. We decided to use a solution containing PBS with 10% ethanol to prepare all the different concentrations of HEPES for analysis, because this solution is also used to dilute the final radiopharmaceutical product.

#### 2.2.2. Specificity

Under the chromatographic conditions described in Section 4, Materials and Methods, the HEPES peaks were well resolved.

In Figure 3, a typical chromatogram of blank eluent is shown in comparison to the spiked samples of HEPES analyzed for the analytical HPLC method validation.

The average retention time of HEPES was 2.440 min ± 0.016.

#### 2.2.3. Linearity

Determination of the linearity of HEPES was carried out on eight sets of solutions in the concentration range of 3–100 μg/mL.

From the evaluated chromatograms, we determined the peak area and graphically plotted these values against the concentrations, thus obtaining the calibration curve. From the regression analysis, a linear equation was obtained (y = 0.029x + 0.098), as well as the correlation coefficient (R^2^) value that was found to be 0.999 in accordance with the acceptance criteria (Table 2), indicating a linear relationship between the concentration of the analyte and area under the peak. The linearity of this method was statistically confirmed for the calibration curve (Figure 4).

For each point of calibration standard, the concentrations of HEPES were recalculated from the equation of the linear regression curve. The average coefficient of variation (CV%) was less than 2%, which is in accordance with the acceptance criteria, as well as the bias% value of less than 5% (Table 2). All data are provided in Appendix A. The limit of quantification (LOQ) and the limit of detection (LOD) for HEPES, calculated according to ICH Q2 (R1) recommendations [16], were 1.69 µg/mL and 0.56 µg/mL, respectively.

#### 2.2.4. Accuracy and Precision of HPLC Analytical Method: Intra-Day–Inter-Day

The method was validated by evaluating the intra- and inter-day precision.

Appendix A show the average values of the concentrations that were recalculated on the basis of the calibration curve; the CV%; the precision index of the collected data, which must be ≤2%; and the accuracy, expressed as bias% ([average concentration observed/nominal concentration × 100] − 100), which must not deviate more than 5% for all the concentrations.

The method was considered to be precise in obtaining the coefficients of variation between 0.09% and 1.99% for intra-day and 0.03% and 0.23% for inter-day precision, indicating that this method presents a good precision generally.

### 2.3. HPLC Analysis of Residual HEPES Content in Different [^68^Ga]Ga-Radiopharmaceuticals

The residual HEPES content in the final formulations of [^68^Ga]Ga-DOTATOC; [^68^Ga]Ga-PSMA-11; [^68^Ga]Ga-Pentixafor; and [^68^Ga]Ga-NODAGA-Exendin-4 that were analyzed by the developed HPLC method are shown in Figure 5.

To ensure that the HEPES peak was not altered by the peak of the radiopharmaceutical, an injection of the cold radiopharmaceutical without HEPES was performed (Figure 6).

Moreover, the residual content concentration of HEPES was calculated using the calibration curve equation that was obtained (y = 0.029x + 0.098).

The results are shown in Table 3. They confirmed that the amount of residual HEPES was well below 200 μg/V_injected_ in all the ^68^Ga-labelled radiopharmaceuticals.

## 3. Discussion

Since HEPES is considered to be a chemical impurity in radiopharmaceutical preparations, an ad hoc TLC method has been introduced in the Ph. Eur. Monographs of [^68^Ga]Ga-radiopharmaceuticals [28,29].

However, it has been proven that the physical separation of the components by TLC, as well as their coloring through the exposure under iodine beads, is neither reliable nor suitable to detect the presence of residual HEPES [31,32,33,34], indicating the need to improve this method or develop a new one.

Moreover, the low limit of HEPES residual content that was imposed by Ph. Eur. (200 μg/V_injected_), and the lack of toxicological data—especially after intravenous administration [32,35,36]—has hampered the use of HEPES in gallium-68 radiopharmaceuticals, despite its superior buffering properties.

In the present study, we describe a new HPLC method to quantify residual HEPES present in [^68^Ga]Ga-radiopharmaceuticals.

We have demonstrated that the use of various solvents (ammonium formate, PBS, PBS + 10% EtOH) does not affect the quantification of HEPES with our developed HPLC method, affording it a particular advantage when various [^68^Ga]Ga-radiopharmaceuticals are formulated with different solutions.

Moreover, the method resulted in greater sensitivity and reliability, not only compared to the Ph. Eur. TLC method, but also with respect to other published HPLC methods [31,34], showing a LOQ of 3 μg/mL. The analysis of different [^68^Ga]Ga-radiopharmaceuticals batches ([^68^Ga]Ga-DOTATOC; [^68^Ga]Ga-PSMA-11; [^68^Ga]Ga-DOTATATE; [^68^Ga]Ga-Pentixafor; and [^68^Ga]Ga-NODAGA-Exendin-4) showed that the amount of HEPES was well below the limit that was described in Ph. Eur., according to the acceptance criteria. The lowest amount was detected in [^68^Ga]Ga-NODAGA-Exendin-4. The difference is most likely due to the different labeling methods that require different volumes and concentrations of HEPES, as well the purification method that is used for [^68^Ga]Ga-NODAGA-Exendin-4, compared to the other radiopharmaceuticals.

Using only the TLC method, it cannot be said that in [^68^Ga]Ga-NODAGA-Exendin-4, the residual quantity of HEPES is less than in the other radiopharmaceuticals, even if it does not exceed the maximum limit.

Indeed, a quantitative method like our developed HPLC method would assess not only the presence of residual HEPES, as did TLC, but it would also precisely quantify the HEPES amount at much lower levels than the maximum limit of 200 μg/V_injected_).

In addition, this new HPLC method enables the assessment of HEPES content within 4 min, because its retention time is 2.440 min and therefore suitable when a pre-release is required, as described in the Ph. Eur.

The validation of the UV-Radio-HPLC method showed an excellent linearity between 3 and 100 μg/mL for HEPES, with an R2 value for calibration curves that was equal to 0.999, CV% < 2%, and a deviation value of bias% from 100% to 5% for all the concentrations.

The precision of this method was statistically confirmed for both intra-day and inter-day assessment.

## 4. Material and Methods

### 4.1. Reagents and Solvents

All the chemicals that were used for the radiolabelling reaction (saline, ethanol, HEPES buffer solution, and water) were of the highest available purity grade and commercially obtained as a single disposable kit (reagents and cassettes for the synthesis of ^68^Ga-peptides using cationic purification ABX, Advanced Biochemical Compounds, Radeberg, Germany).

The GMP peptides DOTATOC, DOTATATE, and PSMA were purchased as lyophilized powder from ABX, Pentixafor from PentixaPharm GmbH (Würzburg, Germany) and NODAGA-exendin-4 from piCHEM (Forschungs- und Entwicklungs Grambach, Austria).

Ga-68 (t1/2 = 68 min, β+ = 89%, and EC = 11%) was obtained from a pharmaceutical grade ^68^Ge/^68^Ga generator (1850 MBq, GalliaPharm^®^, Eckert & Ziegler, Berlin, Germany) by elution with 0.1M HCl (Rotem GmbH, Leipzig, Germany). The amount of detected metal impurities as provided by the manufacturer was less than the defined limit in the European Pharmacopeia monograph [29,37].

The reagents ammonium formate and ammonia were used for UV-Radio-HPLC, as well as acetonitrile and metal-free water, all of which were purchased from Sigma Aldrich (Saint Louis, MO, USA).

Iodine beads were obtained from the Department of Chemistry, Life Science and Environmental Sustainability of Parma University, and silica gel 60 F245 (20 × 20 cm) TLC plates from Merck (Taufkirchen, Germany).

Stock reference solutions (250 μg/mL) and appropriate dilutions of HEPES were prepared in ultrapure water (Sigma Aldrich) and stored at −20 °C.

The aseptic production was conducted in a GMP grade A hot cell (NMC Ga-68, Tema Sinergie). Both the ^68^Ge/^68^Ga generator and the automated synthesis module (Scintomics GRP^®^ module, Germany) were placed in the hot cell.

### 4.2. [^68^Ga]Ga-Radiopharmaceuticals

The [^68^Ga]Ga-radiopharmaceuticals were synthesized using a fully automated platform for labelling synthesis with a disposable cassette system (SC-101 and SC-102, ABX).

Reaction parameters such as reaction time, temperature, and radioactivity were monitored in real time. The process included the pre-concentration of the generator elute through a strong cation exchange (SCX) cartridge, pre-conditioned with 10 mL 0.1 M HCl solution. The same solution was used for the recovery of Ga-68 (III) from the SCX cartridge into the reactor.

The reaction mixture contained each peptide, previously dissolved with 1.5 M or 2.5 M 4-(2-hydroxyethyl)-1-piperazineethanesulfonic acid (HEPES) buffer solution, and then incubated in the heating block for 15 min at 95–100 °C.

After the reaction completion, the crude product was cooled down and purified with a standard purification technique, a solid-phase extraction using cartridges (Sep Pak C18 RP or hydrophilic–lipophilic balance (HLB). Purification was followed by washing steps, and finally, by the elution of the product into a sterile 25 mL vial. The buffer should be completely removed during this procedure; in reality, the used separation techniques are rarely quantitative, so the remaining buffer content has to be assessed during the QC phase, prior to patient administration

### 4.3. HEPES Buffer and Its Evaluation in [^68^Ga]Ga-Radiopharmaceuticals according to Ph. Eur.

For complexation with radiometals, the pH value is one of the most important parameters to be evaluated and optimized.

At a low pH, the coordinating moieties of a chelator can be protonated and therefore inactivated. Focusing on Ga-68, the pH must be in a range of 3.5 to 5.5 because Ga-68 starts to form insoluble colloids and hydroxides at a higher pH that are not available for complexation. Correspondingly, pH buffering systems are necessary for a successful product formation. Appropriate buffers for ^68^Ga-labeling are supposed to be weak gallium-complexing agents that hinder hydrolysis to hydroxides and keep the metal in its active ionic state. Sodium acetate (NaOAc); sodium succinate; and 4-(2-hydroxyethyl)-1-piperazineethanesulfonic acid (HEPES) have been reported as the most favorable buffers for ^68^Ga-labeling so far. Nevertheless, the latter was reported to result in better radiochemical yields for [^68^Ga]Ga-radiopharmaceuticals at low precursor amounts. These buffers are reported to have low metallic complexation properties, making them suitable to adjust pH during labelling and reduce the formation of colloids [31].

HEPES, a zwitterionic buffer, listed as a Good’s buffer with a pKa of 3.0–7.55 [38], provided optimum results in terms of molecular activity, reproducibility, reliability, and versatility activity in radiolabeling bioconjugates with Ga-68 [31,35,36,39]. Moreover, it has a high water solubility, making it helpful for 68Ga-labeling in aqueous media.

Hence, HEPES should be the buffer of choice for clinical routine productions and especially scientific labeling approaches [32,33].

Ph. Eur. defines HEPES as a chemical impurity; thus, an ad hoc test has been introduced and a strict limit of 200 μg per maximum administration volume (V) (maximum recommended dose in milliliters) stipulated for the HEPES content in radiopharmaceuticals [29,30].

The ad hoc test is a TLC method that is performed by spotting the radiopharmaceutical product solution next to a reference solution of HEPES (200 μg/V_Injected_) on a plate of silica gel F254 as the stationary phase, and a mixture of water and acetonitrile (25:75 *v*/*v*) as the mobile phase. After the chromatographic run is completed, the plate is subjected to iodine vapors for 4 min. The color intensity of the two spots is compared to verify that the spot related to the radiopharmaceutical is less intense than the one related to the reference solution. This comparison is necessary to determine if the HEPES content within the radiopharmaceutical does not exceed the maximum reference dose (200 μg/V_Injected_).

### 4.4. Development of a HPLC Method for HEPES Quantification

The physical separation of chemical and radiochemical impurities is obtained with Radio-UV-HPLC, which provides a precise and accurate quantification based on the area counts of peaks.

For the development of a feasible UV-HPLC method to detect HEPES, its chemical and physical properties need to be considered, such as a UV/vis absorption wavelength that is below 200 nm, or 220 nm for [^68^Ga]Ga-radiopharmaceuticals [21,22,23,24,25,26]. We started from a ‘‘mother’’ solution with the highest concentration (250 μg/mL) of HEPES, obtained by dissolving the analyte in phosphate buffer solution (PBS) with 10% ethanol and serial dilutions (100, 80, 60, 40, 20, 10, 5, 3 μg/mL).

A HPLC analysis was performed on a Dionex Ultimate 3000 HPLC system (Thermo Fisher Scientific, Waltham, USA) that was equipped with a Waters Xbridge^®^ column C18 (150 mm × 4.6 mm, 3.5 μm). This was connected to a UV detector set to a wavelength of 195 nm, and a γ-detector (Berthold Technologies, Milan, Italy). The mobile phase consisted of ammonium formate 20 mM pH 9.5 at an isocratic flow of 0.7 mL/min (Table 4).

Chromatograms were collected and analyzed with Chromeleon 7 software^®^ system.

### 4.5. Validation of HPLC Analytical Method

Validation of the analytical method for the quantification of HEPES was carried out according to ICH Q2 (R1) guidelines and EDQM guidelines [16,17,18].

The parameters assessed for the validation were: specificity, linearity, precision (repeatability), accuracy, and limit of quantification. The acceptance criteria for each parameter are listed in Table 2.

#### 4.5.1. Solvent Effect

HEPES mother solution (250 μg/mL) was prepared in either 10% (*v*/*v*) ethanol in PBS or ammonium formate 20 mM, in order to evaluate the matrix effect in the determination of HEPES in solution by our developed HPLC method.

#### 4.5.2. Linearity and Calibration Curve

The determination of linearity was carried out on sets of standard solutions with different concentration for HEPES (100, 80, 60, 40, 20, 10, 5, 3 ug/mL), starting from a ‘‘mother’’ solution with the highest concentration (250 μg/mL). The statistical function used is linear regression with least squares. The curve equation, the correlation coefficient, and the determination coefficient (R^2^) are calculated through this equation: *y = ax + b*, where *y* is peak area, *a* is the slope, *x* is the analyte concentration, and *b* is the intercept [22].

#### 4.5.3. Precision and Accuracy

The precision and accuracy of the method were determined from three validation runs across three separate days with QC samples (*n* = 5 for each run) that were prepared at low, mid, and high concentrations, representative of the validation range.

The accuracy value is expressed as the ratio % between the value of the determined HEPES concentration and the original standard know concentration.

Precision may be considered at different levels as a measure of repeatability or intermediate precision. Repeatability may be calculated based on the content of standard HEPES; the statistical parameter of concern is the variation coefficient (CV%) or relative standard deviation (RSD), which is determined using the equation: CV% = s/m × 100, where *m* is the average of the concentrations and *s* is the standard deviation. The accuracy is a measure of the degree of conformity of a value that is generated by a specific procedure to the assumed or accepted true value, which is performed through bias% value, measuring the difference between the experimental value (calculated from replicate measurements) and the nominal (reference) values. The acceptance criteria is bias% > 95%. The data were obtained by a replicate analysis of samples containing known amounts of analyte at three different concentrations, which were chosen so that they were representative of the validation interval. Five replicates were performed for each concentration for three consecutive days.

Intra-day and inter-day precision are expressed through the coefficient of variation, which must be ≤2% for all concentrations; the intra-day and inter-day accuracy are expressed through the bias%, which must not deviate more than 5% for all the concentrations.

#### 4.5.4. Limit-of-Detection (LOD) and Limit-of-Quantification (LOQ)

The experimental LOQ was determined by analyzing a series of diluted solutions of standard HEPES until a concentration level that was quantified with a precision of >95% was reached. The experimental value (determined as described above) must be confirmed through a precision analysis, using a sample at the concentration corresponding to the found LOQ. The acceptance criteria is CV% < 5%.

Moreover, the LOD and LOQ that were calculated for HEPES were determined based on the standard deviation of the response and the slope, using the following formulas:DL = 3.3 × σ/S
QL = 10 × σ/S
where:

S = calibration curve slope

σ = standard deviation of the response

The two parameters σ and S were assessed with a calibration curve that was created by analyzing four analytes (20-10-5-3 ug/mL) in the range concentration of 3–20 ug/mL—the range in which LOQ is expected to be [16].

### 4.6. Analysis of Residual HEPES Content in Different [^68^Ga]Ga-Radiopharmaceuticals

After the development and validation of the HPLC method for HEPES quantification, we analyzed [^68^Ga]Ga-radiopharmaceuticals ([^68^Ga]Ga-DOTATOC; ([^68^Ga]Ga-DOTATATE; [^68^Ga]Ga-PSMA-11; [^68^Ga]Ga-Pentixafor; and [^68^Ga]Ga-NODAGA-Exendin-4), in order to assess the residual HEPES content in each produced batch.

## 5. Conclusions

The TLC method that was indicated in the Ph. Eur. to assess HEPES impurity in radiopharmaceuticals proved to be insufficiently sensitive and only qualitative. The presented HPLC method was very sensitive, reproducible, and able to exactly quantify the residual HEPES content in all the 68Ga-based radiopharmaceuticals.

Therefore, we developed a rapid, reproducible, and suitable HPLC method for the quantification of HEPES in [^68^Ga]Ga-radiopharmaceuticals, allowing a more complete quality control for release, and guaranteeing patient safety.

## Figures and Tables

**Figure 1 molecules-27-04477-f001:**
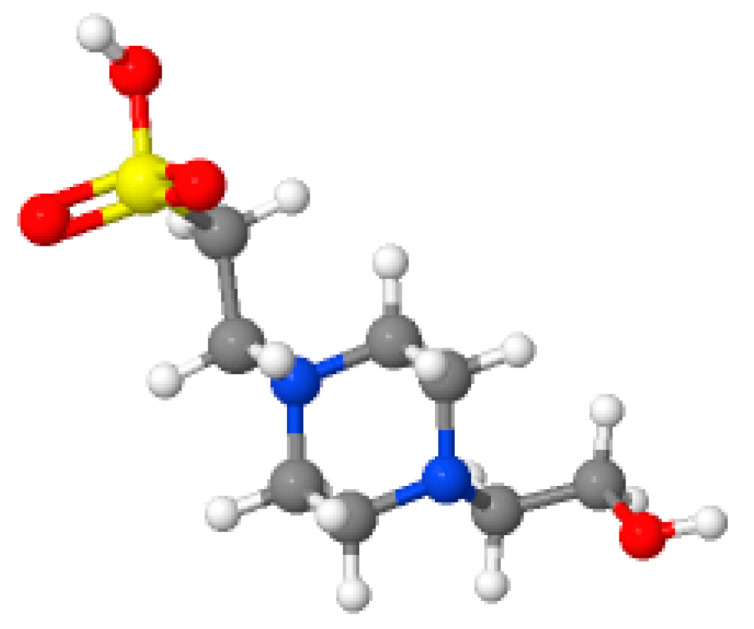
Chemical structure of HEPES (4-(2-hydroxyethyl)-1-piperazineethanesulfonic acid).

**Figure 2 molecules-27-04477-f002:**
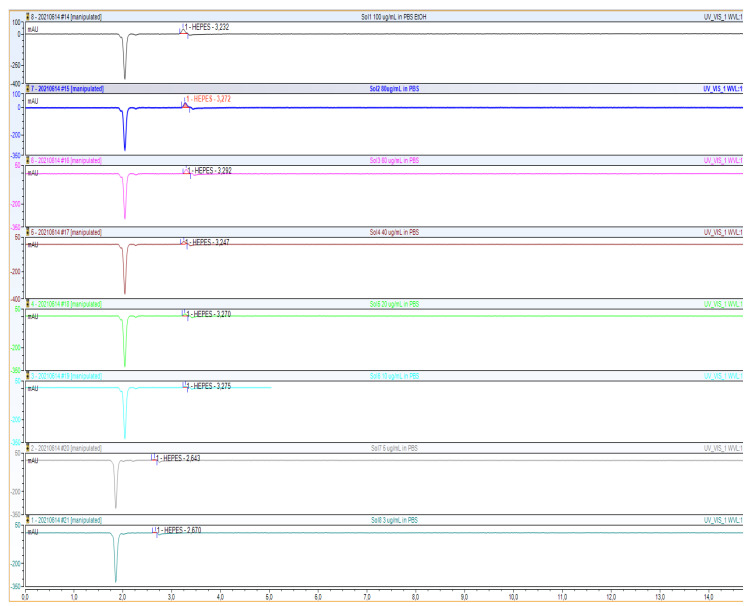
Overlay of the HPLC chromatograms of the analyzed HEPES solutions (100, 80, 60, 40, 20, 10, 5, 3 ug/mL).

**Figure 3 molecules-27-04477-f003:**
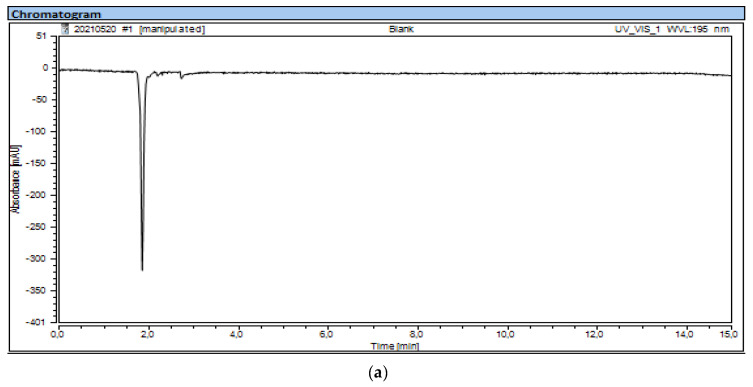
HPLC chromatograms: (**a**) blank; (**b**) sample of HEPES solution.

**Figure 4 molecules-27-04477-f004:**
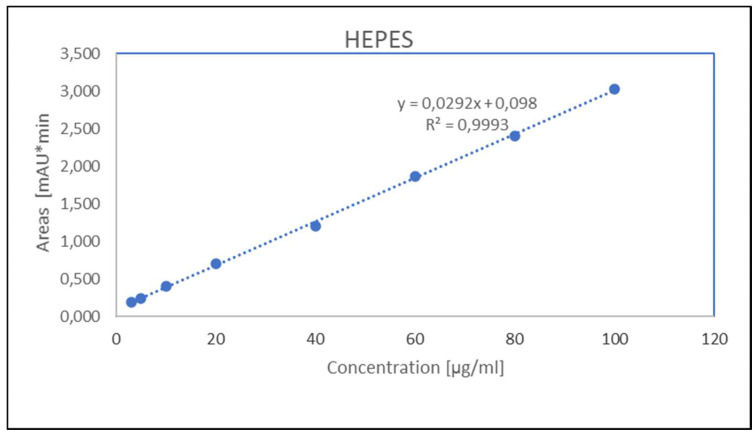
Calibration curve obtained with the average values of peak areas of five different concentrations (100, 80, 60, 40, 20, 10, 5, 3 ug/mL) of HEPES.

**Figure 5 molecules-27-04477-f005:**
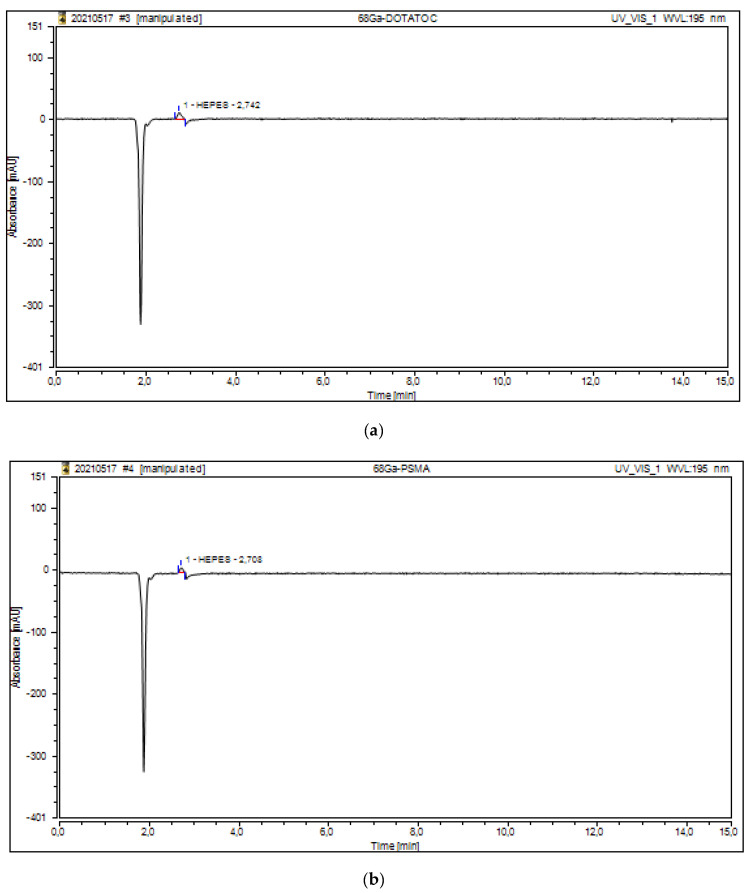
HPLC chromatogram of HEPES content in [^68^Ga]Ga-DOTATOC (**a**); [^68^Ga]Ga-PSMA (**b**); [^68^Ga]Ga-NODAGA-Exendin-4 (**c**); and [^68^Ga]Ga-Pentixafor (**d**).

**Figure 6 molecules-27-04477-f006:**
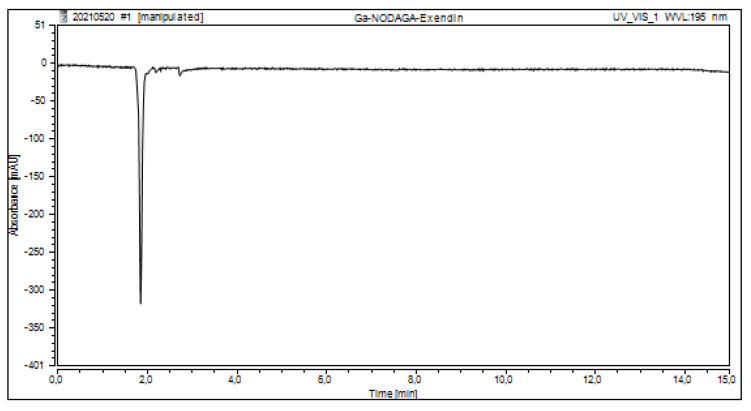
HPLC chromatogram of “cold” Ga-NODAGA-Exendin-4 to assess the content of HEPES in the final radiopharmaceutical.

**Table 1 molecules-27-04477-t001:** Effect of different solvents on HEPES retention time.

Solvent	Retention Time	Concentration of HEPES
Ammonio formate (20 mM)	2.548	100 μg/mL
PBS	2.433	100 μg/mL
PBS + EtOH 10%	2.440	100 μg/mL

**Table 2 molecules-27-04477-t002:** Tests and acceptance criteria in determining chemical purity using HPLC.

Test	Acceptance Criteria
Specificity	≥2.5
Linearity	R^2^ ≥ 0.99
Repeatability	CV% < 2%
Quantification limit (LOQ)	CV% < 5%
Accuracy	Bias% > 95%

**Table 3 molecules-27-04477-t003:** Residual content concentration of HEPES in [^68^Ga]Ga-Radiopharmaceuticals.

Radiopharmaceuticals	Concentration of HEPES	Limit of HEPES
[^68^Ga]Ga-DOTATOC	8.21 μg/mL	12.5 μg/mL
[^68^Ga]Ga-DOTATATE	8.21 μg/mL	12.5 μg/mL
[^68^Ga]Ga-PSMA	10.41 μg/mL	12.5 μg/mL
[^68^Ga]Ga-Pentixafor	9.45 μg/mL	12.5 μg/mL
[^68^Ga]Ga-NODAGA-Exendin	6.45 μg/mL	12.5 μg/mL

**Table 4 molecules-27-04477-t004:** HPLC parameters and values for the quantification of HEPES. The UV-HPLC method was validated according to the regulatory requirements [16,17,18].

Parameters	Value
Stationary phase (column)	Waters Xbridge^®^ column C18 (150 mm × 4.6 mm, 3.5 μm)
UV-Vis (λ)	195 nm
Flow rate	0.7 mL/min
Oven temperature	30 °C
Mobile phase A	Ammonium formate
Time run	15 min
Isocratic	100% A

## Data Availability

All data generated and analysed during this study are included in this published article.

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
