# Peer review of "A Specific HPLC Method to Determine Residual HEPES in [68Ga]Ga-Radiopharmaceuticals: Development and Validation"

_molecules, 2022, doi:10.3390/molecules27144477_

Round 1
Reviewer 1 Report
The manuscript submitted for assessment deals with the topic of determining the residual concentration of HEPES buffer in Gallium-labeled radiopharmaceuticals. The thematic focus corresponds to the focus of the journal, although Pharmaceuticals might be more appropriate. The topic is current and necessary. The authors try to proceed methodically and clearly describe each step performed. However, the level of writing the work shows signs of haste. Some formulations are not happily chosen, eg that nuclear medicine strives in clinical applications for the preparation of quality radiopharmaceuticals with high purity and yield - it always strives for that! There are quite a few typos, typographical and formatting errors in the text. The authors obviously did not check the text in a hurry and should focus on eliminating these errors. This has a very disruptive effect on the overall impression of the work.
The authors describe that in the TLC analysis, the iodine vapor was not detected according to the publication when visualizing the spots belonging to HEPES. So what was observed? Have other detection reagents been tested?
Are the tables in appropriate style? Table 2 - description is placed under table instead above placed one in previous case.
All listed chromatograms are in really poor resolution. Should be improved. In almost all chromatograms is posted [manipulated] - What does it mean?
Could you explaine how was SCX pretreated? line 268
Could you specify type and particle size of silica gel? line 306
It is hard to believe, that authors were used HPLC in combination with gamma detector, as they listed in line 326. Could you specify the type? And how was calibrated? Also information about wavelength might appear earlier. Could you use DAD detector instead of UV? What was the reason of use radiometric detector? I really miss the radiometric data. Did you also compare a realistic samples of radiopharmaceuticals? Could you comment it?
I have several comments and hints should be considered and corrected:
-indexes (e.g. 68Ga) should be in superscript - many cases in the text
- nomenclature of labeled compounds must be unified and written in same style. I personally recommend to use IUPAC style (lines 99 and 100 ought to be checked)
- Dot at the beginning of the line 144
- line 166 - must be corrected
- missing brackets Table 3 - pentixafor
- line 197, ad hoc might be in italic style
- line 204, Vinjected, same in line 228
- line 209, should be ammonium formate
- line 239 - 241 out of style
- missing numbers of equations - between line 378 - 379
- references should be put in proper format without any colours
Overall, the scientific content is at a very good level and can be accepted after corrections. Errors in the text must be corrected and great attention must be paid to the preparation of the manuscript and to avoid hasty writing. I recommend accepting the manuscript after making corrections.
Author Response
Response to Reviewer 1 Comments
Point 1: The authors describe that in the TLC analysis, the iodine vapor was not detected according to the publication when visualizing the spots belonging to HEPES. So what was observed? Have other detection reagents been tested?
Response 1:
After 4 min incubation (according to the Ph Eur) in the iodine vapor chamber, none of the 68Ga-radiopharmaceutical spots as well as HEPES reference solution showed a (bright) yellow signal.
Any other reagent has been tested.
Point 2: Are the tables in appropriate style? Table 2 - description is placed under table instead above placed one in previous case.
Response 2:
Tables are ok and I have corrected the description of table 2, putting it above, as previous cases.
Point 3: All listed chromatograms are in really poor resolution. Should be improved. In almost all chromatograms is posted [manipulated] - What does it mean?
Response 3:
I tried to improved the resolution of the chromatograms.
''Manipulated''posted in the chromatograms means that peaks were integrated and resolved or if there were automated and wrong integrations, they were calncelled.
Point 4: Could you explaine how was SCX pretreated? line 268
Response 4:
SCX pretreated means that the cartrodge was pre-conditioned with 10 mL 0.1 M HCl solution, as explained at page 11/16 in paragraph 4.2.
Point 5: Could you specify type and particle size of silica gel? line 306
Response 5:
Plate of silica gel F254 is a type of stationary phase in which the particle size is almost 5 to 20 µm.
Point 6: It is hard to believe, that authors were used HPLC in combination with gamma detector, as they listed in line 326. Could you specify the type? And how was calibrated? Also information about wavelength might appear earlier. Could you use DAD detector instead of UV? What was the reason of use radiometric detector? I really miss the radiometric data. Did you also compare a realistic samples of radiopharmaceuticals? Could you comment it?
Response 6:
We use a Radio-UV-HPLC (Dionex Ultimate 3000 HPLC system, Thermo Fisher Scientific) that is connected with a gamma-detector (Berthold Technologies, Milan, Italy) an UV detector set to a wavelength of 195 nm in order to detect the signal coming from the absorption of the analysed molecule (HEPES). The UV wavelength chosen (195 nm) is the optimal one seen HILIC structure and its possible related impurities in that mobile phase system. In general, this wavelength can detect more easily (ie higher sensitivity) compounds which have no (UV) chromophore groups such as HEPES analyte.
We do a calibration of the instrument every 6 months using a standard solution with a known concentration.
We don’t have a DAD detector, therefore we use only UV and gamma-detectors.
We need to use a radiometric detector because the radiopharmaceuticals are molecules composed of a radionuclide (radioactive signal detected by the gamma-detector) and a molecular structure (UV absorption detected by UV detector). HEPES is not a radioactive substance, therefore in this paper there are no radiometric data about it, but only data about the signal of molecule absorption, coming from the UV detector, using the same radio-UV-HPLC used for radiopharmaceutical analysis. We compared the realistic samples of radiopharmaceuticals, as shown in figure 5, analysing them also with UV-detector, in order to detect HEPES residual.
Point 7: I have several comments and hints should be considered and corrected:
- indexes (e.g. 68Ga) should be in superscript - many cases in the text: I have corrected it in all the Manuscript
- nomenclature of labeled compounds must be unified and written in same style. I personally recommend to use IUPAC style (lines 99 and 100 ought to be checked): I have corrected it in all the Manuscript.
- Dot at the beginning of the line 144: I have corrected it in all the Manuscript.
- line 166 - must be corrected: I have corrected it in all the Manuscript.
- missing brackets Table 3 – pentixafor: I have corrected it in all the Manuscript.
- line 197, ad hoc might be in italic style: I have corrected it in all the Manuscript.
- line 204, Vinjected, same in line 228: I have corrected it in all the Manuscript.
- line 209, should be ammonium formate: I have corrected it in all the Manuscript.
- line 239 - 241 out of style: I have corrected it in all the Manuscript.
- missing numbers of equations - between line 378 – 379 : I have corrected it in all the Manuscript.
- references should be put in proper format without any colours: I have corrected it in all the Manuscript.

Reviewer 2 Report
The authors present the validation of an analytical methodology by HPC for the quantification of HEPES in 2[68Ga]Ga-Radiopharmaceuticals. The relevance of the research is high, bearing in mind that the analytical methodology presented by the pharmacopoeia (thin layer chromatography) is quantitative, in addition to being unreliable, a factor that was verified by the authors, by demonstrating that some concentrations detected with HPLC do not were detected with TLC.
In this context, the method presented by the authors is a more reliable alternative than the one presented by the regulations (pharmacopoeia).
It is recommended to accept the manuscript after making some major changes.
Although the authors show that the proposed methodology (HPLC), possibly is better, the results presented are not supported by a solid statistical analysis.
Example: the correlation coefficient is not sufficient proof to indicate linearity, for example, an analysis of variance (ANOVA) must be carried out, and homoscedasticity evaluated by applying the G test of the Cochran test (a statistical test can be applied different). The foregoing in relation to each of the factors evaluated
Other recommendations to consider are:
1. Stability analysis
2. Determination of maximum lambda (195 nm). Is important to describe the technical reasons for performing the analysis at this wavelength. (it is recommended to present spectrum).
Author Response
Response to Reviewer 2 Comments
Point 1: Although the authors show that the proposed methodology (HPLC), possibly is better, the results presented are not supported by a solid statistical analysis.
Example: the correlation coefficient is not sufficient proof to indicate linearity, for example, an analysis of variance (ANOVA) must be carried out, and homoscedasticity evaluated by applying the G test of the Cochran test (a statistical test can be applied different). The foregoing in relation to each of the factors evaluated.
Response 1:
- The evaluation of homoscedasticity was carried out performing G Cochran test. We used five replicates of standard solutions at eight different concentration levels. After estimating the variances of the eight data sets, the value of G was calculated as:
G = S2max/SS2
This was compared with the tabulated value at the significance level of α 0.05, for eight different concentration levels (8 variances) and 5 replicates.
G0.95; 5; 8 = 0.3910
The experimental value (G = 0.3585) was lower than the tabulated one, excluding that the used method was homoscedastic.
conc [μg/ml] |
aree |
variances |
S variances |
G Cochran |
|
|
|||||
100 |
3,032 |
1,043E-06 |
|
0,0003474 |
0.35848 |
3,033 |
|
||||
3,033 |
|
||||
3,035 |
|
||||
3,034 |
|
||||
80 |
2,410 |
1,25E-04 |
|
||
2,423 |
|
||||
2,395 |
|
||||
2,410 |
|
||||
2,400 |
|
||||
60 |
1,867 |
5,21E-05 |
|
||
1,860 |
|
||||
1,880 |
|
||||
1,868 |
|
||||
1,870 |
|
||||
40 |
1,200 |
1,83E-05 |
|
||
1,205 |
|
||||
1,209 |
|
||||
1,211 |
|
||||
1,208 |
|
||||
20 |
0,699 |
9,08E-05 |
|
||
0,720 |
|
||||
0,707 |
|
||||
0,695 |
|
||||
0,705 |
|
||||
10 |
0,397 |
3,79E-05 |
|
||
0,398 |
|
||||
0,409 |
|
||||
0,409 |
|
||||
0,409 |
|
||||
5 |
0,239 |
1,88E-05 |
|
||
0,247 |
|
||||
0,239 |
|
||||
0,246 |
|
||||
0,240 |
|
||||
3 |
0,184 |
3,81E-06 |
|
||
0,189 |
|
||||
0,187 |
|
||||
0,186 |
|
||||
0,186 |
|
- Five calibration lines were set up at eight different concentration levels (100 -3 µg / ml). The data obtained were analyzed with the GraphPad software Prism v. 9. We employed a global regression approach and used an F test, implemented in the software, to compare a global model in which slope and intercept were the same among data sets with each single model and its own slope and intercept. The null (H0) hypothesis was that one global model could fit all datasets. The alternative hypothesis (H1) was to employ different curves for each dataset. The test yielded an F value of 0.05464 (8, 30) and based on an alpha value of 0.05 the H0 hypothesis was not rejected.
In the next page, the table of results:
aree |
aree |
aree |
aree |
aree |
Global (shared) |
|
Comparison of Fits |
||||||
Null hypothesis |
One curve for all data sets |
|||||
Alternative hypothesis |
Different curve for each data set |
|||||
P value |
0,9999 |
|||||
Conclusion (alpha = 0.05) |
Do not reject null hypothesis |
|||||
Preferred model |
One curve for all data sets |
|||||
F (DFn, DFd) |
0,05464 (8, 30) |
|||||
Different curve for each data set |
||||||
Best-fit values |
||||||
YIntercept |
0,09390 |
0,1007 |
0,1015 |
0,09879 |
0,09961 |
|
Slope |
0,02913 |
0,02917 |
0,02905 |
0,02917 |
0,02910 |
|
95% CI (profile likelihood) |
||||||
YIntercept |
0,05118 to 0,1366 |
0,06052 to 0,1409 |
0,05502 to 0,1479 |
0,06296 to 0,1346 |
0,05799 to 0,1412 |
|
Slope |
0,02831 to 0,02994 |
0,02841 to 0,02994 |
0,02817 to 0,02993 |
0,02849 to 0,02985 |
0,02831 to 0,02990 |
|
Goodness of Fit |
||||||
Degrees of Freedom |
6 |
6 |
6 |
6 |
6 |
|
R squared |
0,9992 |
0,9993 |
0,9991 |
0,9995 |
0,9993 |
|
Sum of Squares |
0,006274 |
0,005555 |
0,007417 |
0,004413 |
0,005956 |
|
Sy.x |
0,03234 |
0,03043 |
0,03516 |
0,02712 |
0,03151 |
|
One curve for all data sets |
||||||
Best-fit values |
||||||
YIntercept |
0,09889 |
0,09889 |
0,09889 |
0,09889 |
0,09889 |
0,09889 |
Slope |
0,02912 |
0,02912 |
0,02912 |
0,02912 |
0,02912 |
0,02912 |
95% CI (profile likelihood) |
||||||
YIntercept |
0,08515 to 0,1126 |
0,08515 to 0,1126 |
0,08515 to 0,1126 |
0,08515 to 0,1126 |
0,08515 to 0,1126 |
0,08515 to 0,1126 |
Slope |
0,02886 to 0,02939 |
0,02886 to 0,02939 |
0,02886 to 0,02939 |
0,02886 to 0,02939 |
0,02886 to 0,02939 |
0,02886 to 0,02939 |
Goodness of Fit |
||||||
Degrees of Freedom |
38 |
|||||
R squared |
0,9992 |
0,9993 |
0,9991 |
0,9994 |
0,9993 |
0,9993 |
Sum of Squares |
0,006472 |
0,005692 |
0,007469 |
0,004454 |
0,005960 |
0,03005 |
Sy.x |
0,02812 |
|||||
Constraints |
||||||
YIntercept |
YIntercept is shared |
YIntercept is shared |
YIntercept is shared |
YIntercept is shared |
YIntercept is shared |
|
Slope |
Slope is shared |
Slope is shared |
Slope is shared |
Slope is shared |
Slope is shared |
|
Number of points |
||||||
# of X values |
8 |
8 |
8 |
8 |
8 |
|
# Y values analyzed |
8 |
8 |
8 |
8 |
8 |
Point 2: Stability analysis.
Response 2:
Stability tests of the HEPES analyte were performed on five replicates of the QC concentration after 8h at room temperature (short time stability) and after three freeze-and-thaw cycles. The overall stability showed % nominal values ranging from 96.25 to 98.48 for room temperature short-term-stability and from 96.51 to 98.46 for freeze-and-thaw stability.
Point 3: Determination of maximum lambda (195 nm). Is important to describe the technical reasons for performing the analysis at this wavelength. (it is recommended to present spectrum).
Response 3:
The UV wavelength chosen (195 nm) is the optimal one seen HILIC structure and its possible related impurities in that mobile phase system. In general, this wavelength can detect more easily (ie higher sensitivity) compounds which have no (UV) chromophore groups such as HEPES analyte.
Here the spectrum of HEPES:

Round 2
Reviewer 2 Report
I recommend publishing the manuscript in its current form.